# Improving Adolescents’ Subjective Well-Being, Trait Emotional Intelligence and Social Anxiety through a Programme Based on the Sport Education Model

**DOI:** 10.3390/ijerph16101821

**Published:** 2019-05-23

**Authors:** Pablo Luna, Jerónimo Guerrero, Javier Cejudo

**Affiliations:** Department of Psychology, Faculty of Education of Ciudad Real, Universidad de Castilla-La Mancha, Ronda de Calatrava 3, 13071 Ciudad Real, Spain; pablo.luna@uclm.es (P.L.); jeronimo.guerrero@alu.uclm.es (J.G.)

**Keywords:** physical education, social and emotional learning, sport education model, subjective well-being, trait emotional intelligence, social anxiety

## Abstract

This study aimed to evaluate the impact of a physical-sport education pilot programme on adolescents’ subjective well-being (health-related quality of life, positive affect and negative affect), trait emotional intelligence and social anxiety. The programme was based on the pedagogical sport education model within a quality physical education framework, and approached from the perspective of social and emotional learning. Participants were 113 compulsory secondary education students aged 12–15 years that were assigned to a control group (*n* = 44) and an experimental group (*n* = 69). A quasi-experimental design with repeated pre-test and post-test measures was used. Bonferroni correction was applied for multiple comparisons. The preliminary results obtained in this investigation revealed that the physical-sport education pilot programme promoted significant improvements in a specific indicator of subjective well-being and trait emotional intelligence in the experimental group. These encouraging findings support the pedagogical efficiency of the programme with regard to the programme aim. The findings also highlight the feasibility and appropriateness of the programme in terms of an innovative teaching proposal.

## 1. Introduction

In the education field, physical education is a subject that can contribute to improving the well-being and health of children and adolescents. The concept of quality physical education, which is understood as an interrelated system of inclusive and active teaching and learning, must be considered a key framework for integral approaches (i.e., education and health) [1]. It can also be seen as a physically active teaching and learning experience that can positively impact students’ psychomotor abilities, cognitive comprehension and social and affective aptitudes [2]. Moreover, in our view, quality physical education could be grouped in the social and emotional learning category.

Quality physical education aims to achieve an integral education commitment [3,4] that allows students to be physically literate [5,6]. Physical literacy is the pillar of quality physical education, and can be defined as the motivation and cognitive, physical and affective competence necessary to encourage and preserve an active attitude in life, enabling a positive development of the aptitude to achieve, understand and use decisions about one’s health efficiently [7]. Students who are physically literate intrinsically value their own psychomotor capabilities, as well as the contribution of these abilities to well-being and health [6].

The connection between health and physical activity is widely accepted [8,9]. However, the effects of physical activity on health in the educational context should be deepened through experimental studies. Designing teaching and learning processes in this area will support physical, psychological, emotional and social development [10].

The synergy between the practice of physical-sport activity together with physical and psychological health is a gradually growing interest area for education researchers [11,12,13,14]. Moreover, different investigations in the education framework of the evolution of quality physical education have emphasised the need for methodological change [4,15]. Several pedagogical models share the same features [16].

This study is based on quality physical education, manifested through a specific sport education model [17]. Sport education is a pedagogical model that uses essential features of sports (seasons, competitions, membership, data register, culminating event and festivity), and aims to achieve the inclusive goal of all students living real and meaningful sport experiences in physical education. In addition, this model aspires to develop competence, enthusiasm and a physical-sport culture in students [18].

The pedagogical potential of the sport education model, if correctly implemented [19], results in benefits at a physical level [20,21,22]. Similarly, it has been shown to have a positive impact on psychological variables in adolescents. These positive benefits include: basic psychological needs [23]; improvement in competence [24] and the feeling of belonging to a group [25]; decrease in attitudes towards violence and improvements in social responsibility and participants’ relationships [25,26]; more self-determined behaviour [27]; improvements in friendship and sport goals [24]; decrease in aggressive behaviour and improvements in friendship relationships [25,28]; positive changes in the perception of the social climate [29]; improvement in social relationships [30]; improvement in trait emotional intelligence and motivational mediators [31]; and improvement in sport culture and enthusiasm. However, no benefits in terms of life satisfaction have been found [32]. By making use of sport, these studies provide evidence of a meaningful and positive impact on the psychological and physical development of the school-age population [11,33]. 

Health is generally determined by several physical–biological, psychological and social indicators [34]. Therefore, good health is a fundamental dimension in personal and social progress, and an important sphere in quality of life [35]. Approached from the perspective of positive health, a state of wellness encourages individuals to reach complete social and psychological development [36].

The World Health Organization (WHO) aims to promote physical and psychological health [34] that supports a good quality of life [37]. The construct of subjective well-being is among the factors that affect health. Consequently, research on the influence of subjective well-being in different social and educational contexts has received increasing attention in recent decades [38]. Subjective well-being comprises two main factors: a cognitive aspect (satisfaction with one’s own life) and an affective aspect (positive and negative affect) [39,40]. The cognitive side of well-being reflects the assessment of how individuals process information in their lives [41]. The affective side of well-being implies a hedonistic individual balance; that is, how often individuals experience positive and negative emotions [39,40].

Recent research has focused on studying the effects of positive psychological variables on personal and social development [42]. These studies have been categorised as positive psychology [43]. Emotional intelligence is a positive variable that currently has broad support because of its close connections to subjective well-being and physical and mental health [38,44,45].

Variables such as social anxiety have a negative effect on subjective well-being [46]. In this sense, social anxiety can be defined as a person’s constant fear of one or more social or performance situations, in which they are exposed to unknown persons or the possible scrutiny of other people [47]. Social anxiety has a negative impact on subjective well-being in adolescents because of the anguish individuals may feel [48], which in turn may negatively affect the quality of their interpersonal relationships [49,50].

We consider that education should promote social and emotional learning, which the WHO defines as a heterogeneous set of life skills, as this is a potential factor that supports and encourages mental health [51]. Scholars in favour of this teaching proposal argue that emotional education may also promote public health [52,53], because it’s ultimate goal is improvement of the general quality of health and well-being in citizens. In the school context, many researchers claim that a key purpose of education is to improve peoples’ lives so that they can reach an optimal degree of personal happiness and well-being in adulthood [54]. This suggests that a healthy pedagogical and psychological school environment may facilitate students’ positive adjustment; therefore, such an environment is essential for the development of well-being in children and adolescents [55].

The theoretical and practical justification for this study was rooted in the work of various authors who developed educational interventions based on the sport education model and recommended that further research should evaluate the impact of this model on the promotion of optimal personal and social development [25,26,56,57]. In this sense, Metzler’s [16] contributions are very relevant, stating that a sports model teaching program is mainly focused on different domains [16]: affective, cognitive and motor. In line with this statement, our study focuses on the affective domain. We agree with several previous authors [39,40] that subjective well-being is a key variable that influences balanced personal and social development. In addition, the existing relationship between subjective well-being and trait emotional intelligence suggests that it is necessary to further explore this topic. Social anxiety generates inappropriate social relationships in adolescents [46,48]. Given the positive effects of the sport education model on social relationships [30], it is possible that such interventions may reduce social anxiety.

This study aimed to evaluate the impact of a pilot programme based on the sport education model on the three variables: subjective well-being, trait emotional intelligence and social anxiety. The hypotheses focused on the assumptions that the programme will result in improvements in our participants’ subjective well-being (Hypothesis 1), trait emotional intelligence (Hypothesis 2) and social anxiety (Hypothesis 3).

## 2. Methods

### 2.1. Participants

This study used non-probability incidental or accessibility sampling. The sample comprised 113 students in compulsory secondary education aged 12–15 years (mean age (*M*) = 13.82 years, standard deviation (*SD*) = 0.79 years). The research was conducted in a state school with students from five class groups. The control group comprised 44 students (two class groups) and the experimental group included 69 students (three class groups). The experimental and control group assignment was based on a cluster-randomised controlled trial. The gender distribution was 64 (57%) boys and 49 (43%) girls (Table 1).

The main inclusion criterion was parental consent. The exclusion criteria were: (a) attending less than 80% of the sessions of the intervention programme (less than 13 sessions); (b) students with special educational needs associated with intellectual disability; and (c) students that were removed from school for disciplinary reasons.

### 2.2. Procedure

We requested the collaboration of the educational centre (Spain) in this study. The management board of the participating school was contacted to obtain their approval and authorisation for the study. Permission was also obtained from the families of the participating students, and from the teaching staff and school council. This study respected the relevant ethical values and guaranteed participants’ confidentiality and anonymity. In addition, this study was developed in accordance with the Declaration of Helsinki regarding human experimentation. The study procedures were conducted in accordance with the Universidad de Castilla-La Mancha code of ethics.

This study used a quasi-experimental design with repeated pre-test and post-test measures and a control group. The study was conducted over three stages. First, before the intervention began, the assessment instruments (pre-test evaluation) were handed out for completion in the first 20 minutes of two sessions, to avoid burdening the students. Next, the programme based on the sport education model was implemented. The programme sessions took place during the second term of the school year. Finally, the assessment instruments were completed a second time (post-test evaluation). The independent variable was the intervention programme, and the dependent variables were subjective well-being, trait emotional intelligence and social anxiety.

### 2.3. Measures

Four evaluation instruments were used to assess the variables in this study, under the psychometric parameters of reliability and validity.

The Kidscreen-10 Index [58] was used to assess subjective health-related quality of life and well-being. This 10-item scale was designed for children and adolescents aged 8–18 years. Each item has five response options, ranging from ‘Never’ to ‘Always’ or ‘Not at all’ to ‘Extremely’. The 10 items cover: affective symptoms of depressed mood; cognitive symptoms of disturbed concentration; psychovegetative aspects of vitality, energy and feeling well; and psychosocial aspects correlated with mental health, such as the ability to experience fun with friends or getting along well with others at school. The adapted version of the questionnaire used in this study has adequate internal consistency reliability (*Cronbach’s alpha* = 0.82) and test-retest stability (*r* = 0.73; *ICC* = 0.72) [59].

The Positive and Negative Affect Schedule [60] was used to assess participants’ positive and negative affect. The Spanish version of this scale for children and adolescents was validated by Sandín [61]. This scale comprises 20 items on two dimensions: positive affect and negative affect. Each subscale contains 10 items. The questionnaire is completed by participants based on the way they normally feel and behave. The scale has three response options: ‘Never’ = 1, ‘Sometimes’ = 2, and ‘Many times’ = 3.

We used the Trait Emotional Intelligence Questionnaire Adolescents Short Form (TEIQue-ASF) [62] (adapted into Spanish in its abridged version for teenagers by Ferrando and Serra [63]) to evaluate trait emotional intelligence based on the theoretical model of Petrides and Furnham [64]. The 30 items that make up the TEIQue-ASF are scored on a 7-point Likert scale (1 = ‘Completely disagree’ to 7 = ‘Completely agree’). The general emotional intelligence score of the total scale is obtained by summing the 30 items.

Finally, participants completed the Social Anxiety Scale for Adolescents (SAS-A) [65]. The SAS-A comprises 22 items; 18 items are self-descriptive and four are distracting elements that are not taken into account for the score. The SAS-A contains three subscales: (a) fear of negative evaluation (eight items), (b) anxiety and social avoidance before strangers or new social situations (six items) and (c) anxiety and social avoidance in social situations in general (four items). Responses are on a 5-point Likert-type scale from 1 (‘Never’) to 5 (‘Always’). In addition, a global index of social anxiety (SAS-T) is obtained by summing the scores for the items (excluding neutral items). High scores reflect high levels of social anxiety [65]. The scale was adapted to the Spanish population by Olivares, Ruiz, Hidalgo, García-López, Rosa and Piqueras [66]. Only the SAS-T score was used in this study.

### 2.4. Intervention Programme

The physical-sport programme was completed following the sport education model structure [17]: (1) *season*: lengthy didactic units; (2) *membership*: development of a team spirit and cooperation; (3) *regular competition*: showing technical–tactical abilities; (4) *data register*: giving evidence of and analysing the process that has been followed; and (5) *festivity*: a festive atmosphere. This highlighted other important education aspects such as: cooperative learning; autonomy and personal initiative; positive interdependence; and self-management of responsibility roles in conflict resolution (i.e., referee and coach). This helped to make the sport experience more real and positive, including how students transferred responsibilities by means of organisation roles (i.e., referee and scorer), team roles (i.e., coach and physical trainer) and how sport content was modified when adapted to the students [17].

Hastie and Casey’s guidelines were followed for the design and validation of the programme [19] (p. 423): (a) thoroughly detailed curricular elements; (b) precise certification of the applied model; and (c) an in-depth explanation of the context of the programme. The intervention programme was implemented in the experimental group following sequencing of content and activities in three stages (initial, intermediate and final) over 16 sessions (Table 2).

This pilot programme was developed to reflect the teaching hours of the physical education subject, which covers 16 55 minute sessions (2–3 sessions per week for 6 weeks). The total duration was considered sufficient to analyse the possible effects of the programme on the dependent variables, as indicated by previous research [67].

The educational intervention applied to the experimental group consisted of a didactic unit that used an alternative sport, called ringo [68,69]. Ringo is an alternative, modified and reduced sport of divided court and net. It is played with a hoop (ringo) and a volleyball net. The objective is to score when the ringo falls on the opposite court (Figure 1). In this pilot programme, the application of an alternative sport that was novel and unknown to students meant that everyone started with the same theoretical and practical sports knowledge, and there were few initial differences in their levels of technical–tactical sports skill. Alternative sports are characterised by being motivating, cooperative, socialising and adapted to participants’ characteristics. The selection and organisation of teams (five teams per classroom) was developed by drawing lots. In addition, different responsibility roles were assigned to participating students: player; referee; coach–captain; physical trainer; person responsible for statistics and reports; and member of the discipline and organisation committee. An essential rule in the development of the pilot programme was that all students would actively participate in the programme with the assignment of two roles (player role and another responsibility role). The pilot programme also used various learning and curricular resources (self-designed portfolio, worksheets and reports) that had been used by other authors [70].

For the control group, a didactic unit of traditional collective sport with a conventional teaching style was developed [71]. This traditional teaching model aimed to improve students’ technical motor skills only. In the teaching–learning process, the teacher assumed a managerial role and the students adopted passive individual roles limited to following the directive instructions of the teacher. This intervention consisted of 12 55 minute sessions (two sessions per week for 6 weeks). The first nine sessions were aimed at learning the technical fundamentals of basketball (pot, dribbling, passing, throwing and receiving) through a task assignment teaching style [71]. These traditional sports sessions were based on a 10 minute warm-up, 40 minute main session that included explanations and basketball practice and a 5-minute warm down in which stretching was performed. During these sessions, all tasks were directed by the teacher without students’ participation. The last three sessions were dedicated to team competition.

Two compulsory secondary education teachers, both with advanced degrees in Sports Science participated in this research. The first teacher (with a Master’s of Science in Psychology) participated in the design and implementation of the pilot programme. The second teacher developed an intervention based on the traditional model. Both teachers received a 10 hour training course on the specific theoretical and practical aspects of each teaching model. In addition, supervision and tutoring was provided by a researcher expert in the sports education model and a researcher expert in the traditional education model. This tutoring consisted of: (a) session-by-session analysis during the intervention programmes; (b) telephone conversations and emails to resolve doubts, concerns and problems; and (c) weekly visits to the teaching centre. In these visits, the experts visited the centre randomly, without prior notice, with the objectives of: verifying that there were no gaps between what was planned and what was implemented, and checking that the teaching models were applied with all of their characteristics.

### 2.5. Data Analysis

After fulfilment of the requirements of normality and homoscedasticity was verified, we examined the distribution of the data for a univariate normality analysis. The results showed asymmetry and kurtosis values lower than 1.2. Next, we calculated the reliability coefficients (Cronbach’s alpha, composite reliability, average variance extracted and McDonald’s omega coefficient) to obtain reliability evidence. Then, to determine the impact of the programme, descriptive analyses (mean and SD) and analyses of variance (ANOVA) were performed with the scores collected in the pre-test stage. Subsequently, descriptive analyses and analyses of covariance (ANCOVA) were used with post-test scores to determine the impact of the programme on each of the variables. Bonferroni correction was applied for multiple comparisons. For all analyses, a *p*-value <0.05 was considered to indicate statistical significance. After application of Bonferroni correction, a *p*-value <0.012 was considered significant.

The effect size (*µ*^2^) of the differences was calculated using partial eta-squared [72]. The effect size was analysed based on four ranges: 0–0.009, negligible; 0.010–0.089, low-effect size; 0.090–0.249, medium-effect size; and >0.250, big-effect size [72]. The data were analysed with SPSS version 24.0 (IBM Corp., Armonk, NY, USA).

## 3. Results

### 3.1. Reliability

In this study, we used well-established measures with appropriate psychometric properties (Table 3).

### 3.2. Effects of the Programme

The pre-test MANOVA results did not reveal statistically significant differences between the groups prior to the intervention, Wilks’ Lambda, *Λ* = 0.571, *F* (5, 108) = 0.739, *p* = 0.333, with a small effect size (*η2* = 0.062, *r* = 0.11).

The ANOVA using the pre-test scores (Table 4) revealed no statistically significant differences in any of the dependent variables before the programme began, except for a significantly higher score for trait emotional intelligence in the experimental group compared with the control group. The size of the effect was low for trait emotional intelligence (*µ*^2^ = 0.009). Applying Bonferroni correction showed no significant differences in any of the variables.

Results from the pre-test–post-test MANCOVA revealed significant differences between the two conditions, Wilks’ Lambda, *Λ* = 0.899, *F* (5, 108) = 5.295, *p* = 0.003, with an average effect size (*η2* = 0.267, *r* = 0.32).

Next, we performed ANCOVA for the dependent variables using the post-test scores. To assess the magnitude of these differences, the effect size for each variable was calculated by partial eta-squared (Table 4).

#### 3.2.1. Effects on Subjective Well-Being

There was no significant improvement in post-test health-related quality of life in the experimental group (Table 4). The experimental group did not show a significant increase in positive affect scores after testing. We confirmed a significant decrease in post-test negative affect scores in the experimental group (Table 4), with a medium effect size (*µ*^2^ = 0.123; partial eta-squared).

#### 3.2.2. Effects on Trait Emotional Intelligence

The analysis revealed significant improvements in trait emotional intelligence in the experimental group after the programme, with a medium effect size (*µ*^2^ = 0.241) (Table 4).

#### 3.2.3. Effects on Social Anxiety

There were no significant differences between the experimental and the control groups in SAS-T scores (Table 4).

## 4. Discussion

This study evaluated the impact of a pilot programme based on the sports education model on compulsory secondary education students’ subjective well-being, trait emotional intelligence and social anxiety.

The preliminary results obtained in this study revealed significant improvement of a specific indicator of subjective well-being (NA) in the experimental group after the pilot programme. The experimental group did not show a significant improvement in health-related quality of life when compared with the control group. Our results are consistent with the findings reported in other studies [32], which did not confirm significant benefits for life satisfaction among adolescents following a sport education-based experience. Our findings also indicated that the programme showed a significant decrease in negative affect, improving an indicator of the affective component of subjective well-being (i.e., a decrease in negative emotions) [39,40]. These results partially verify Hypothesis 1, and highlight the importance of further research in this context.

This is consistent with the findings reported in other studies [32], as previous studies established a connection between physical activity and subjective well-being [73]. These findings are also consistent with research that argues that active, inclusive and effective teaching and learning processes applied within a quality physical education framework fosters a motivating school climate in affective and psychological terms [1,74]. Pedagogical and methodological aspects highlighted by this intervention pilot programme (e.g., cooperative learning, a feeling of membership to a team, positive interdependence and self-management or autonomy/use of responsibility roles) could have influenced these results. Furthermore, a motivating school context, enabled by the implementation of the sport education model [31], may also strengthen affective bonding in adolescents [12].

Significant improvement in trait emotional intelligence was observed in the experimental group after the programme, which confirmed Hypothesis 2. These results are also consistent with those reported by other authors [31]. The relationship between trait emotional intelligence and subjective well-being [38,45], as well as that between trait emotional intelligence and physical and psychological health [44], may trigger these improvements in adolescents. This indicates that good trait emotional intelligence promotes positive emotional states and a reduction of negative moods, thereby positively impacting well-being and health [45].

We found no significant improvement in students’ social anxiety, meaning that we could not confirm Hypothesis 3. Although this is similar to the results obtained by different authors for social relationships variables [23,24], our results contradicted the findings of other studies [25,26,28,30]. Further research on the effects of the sport education model is therefore necessary, especially given the theoretical specificity of social anxiety and its incidence in social relationships among adolescents. In addition, social anxiety can present opposing consequences. It can have positive effects on social relationships for some individuals, whereas it can have negative effects on others, characterised by anguish and social avoidance [65].

However, in our opinion this study has been very exhaustive in the evaluation methodology of the intervention program (including the Bonferroni corrections). In this sense, we have not found any study on the effectiveness of the sport education model that uses these statistical corrections. This fact could be influencing the comparison of our results with those obtained in other similar researches on the sport education model as a teaching model.

Despite these promising results, this study had some limitations. First, the sampling procedure was chosen for reasons of convenience and not by random procedures. However, allocating students to either the experimental or control group was performed randomly based on the class group to which they belonged. Second, it would be necessary to increase the sample size to minimise potential biases in the results and increase the generalisability. Third, the instruments used were self-reported, and the results might have been influenced by bias related to social desirability in adolescents. It would be necessary to use high-performance tests or hetero-evaluation to minimise such bias. Similarly, differences in the trait emotional intelligence pre-test scores between the experimental and control groups might have had an impact on our results. Finally, it is necessary to highlight the difficulties encountered when following all of the recommendations for the implementation of the sport education model [19]. Similarly, it would be necessary to include session analysis procedure in order to evaluate if the main principles of the model were followed by the teachers [75].

Several aspects can be suggested regarding future lines of investigation, such as increasing the number of participants and diversifying their sociocultural background. It may also be worthwhile analysing the impact of the programme on other variables, such as academic performance and social and school adjustment. Similarly, to study the effects on depression with the use of biological correlates (HPA markers, cortisol immunitarian parameters, etc.). In addition, it would be interesting to conduct a follow-up evaluation to assess the sustainability of the effects of the programme.

This study presents innovative contributions at both theoretical and practical levels. The theoretical contribution is related to fact that the lack of physical activity can have a harmful effect on individual’s health and is currently an important public health concern [76]. In this respect, United Nations Educational, Scientific and Cultural Organization (UNESCO) [1] emphasised the importance of fostering and promoting active behaviours [6] in all contexts, especially at schools. Consequently, it should be noted that there is a positive connection between health and physical activity: sedentarism is a major risk factor for mortality, which gives rise to concern about the prevalence of sedentarism and socio-educative patterns of inactivity, especially in school contexts. At a practical level, our findings may help teaching staff in their tasks at school, as they provide a tool that may be used in teaching practice. In addition, the findings open up interesting fields of research in terms of the application of sport education, especially in terms of its impact on psychological variables.

## 5. Conclusions

In conclusion, our findings suggest that the pilot programme stimulated some improvement in adolescents’ subjective well-being and trait emotional intelligence, but did not impact social anxiety. Therefore, on the basis of quality physical education and the social and emotional learning approach, the implementation of such programmes is recommended given the possible psychological benefits for adolescents in the educational context. A commitment to sports and other physical-sport activity options within a quality physical education framework, efficiently applied by means of relevant pedagogical models (such as sport education), may play an important role in students’ integral development [77,78].

## Figures and Tables

**Figure 1 ijerph-16-01821-f001:**
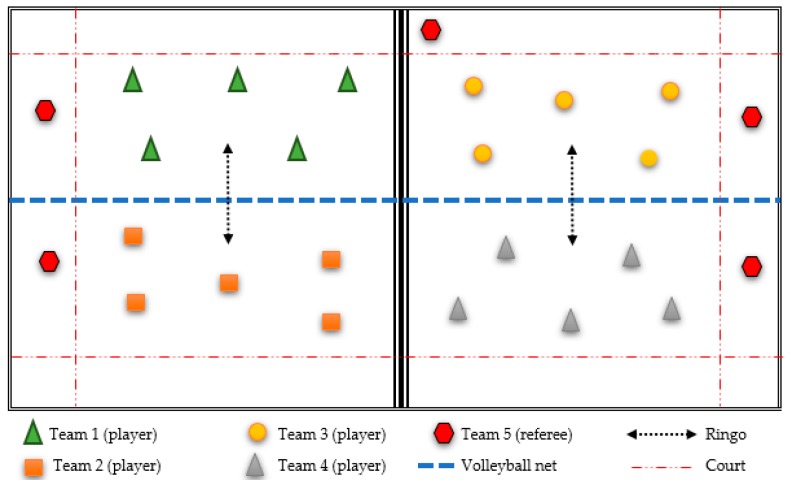
Practical session of the intervention programme.

**Table 1 ijerph-16-01821-t001:** Sex and age of the sample.

		*n*	*%*
**Sex**	Male	64	57
	Female	49	43
**Age**	12	42	37
	13	45	40
	14	23	20
	15	3	3

**Table 2 ijerph-16-01821-t002:** Sequencing of stages and activity sessions in the intervention programme.

Stage	Session	Sport Education Model
Initial (Theoretical Sessions)	1–2	Introduction and presentation of the Sport Education Model with digital and audio-visual support (ICT). Presentation and distribution of learning resources. Division and organisation of classroom groups in teams (assignment of team names with a didactic and cross curricular theme). Distribution and selection of responsibility roles.
	3	Explanation for the self-design of learning resources on digital format (ICT). Selection and assignment of anthems, badges, mascots and t-shirts representing a team.
Intermediate (Practical Sessions)	4–7	Practical implementation of the roles of each member of the teams. Learning of technical-tactical elements and abilities: kicking-off, catching, moving, throwing, defence and attack. Learning game rules.
	8–9	Warming-up, training and friendly matches. Meetings for comprehension and reflection with intervention of the responsibility roles.
	10–14	Regular stage competition (Round Robin).
Final (Practical Sessions)	15–16	Inter-class groups final competitions (final matches with class groups), final event, giving awards and diplomas.

**Table 3 ijerph-16-01821-t003:** Reliability evidence of the instruments used (*n* = 113).

Measures	*α*	*CR*	*AVE*	*Ω*
KIDSCREEN	0.91	0.89	0.674	0.92
PANASN-PA	0.70	0.77	0.502	0.72
PANASN-NA	0.74	0.76	0.519	0.77
TEIQue-ASF	0.71	0.70	0.503	0.79
SAS-T	0.85	0.80	0.687	0.87

Notes: α = Cronbach’s alpha; *CR* = composite reliability, *AVE* = average variance extracted; *Ω* = McDonald’s omega index.

**Table 4 ijerph-16-01821-t004:** Mean, standard deviation, analysis of variance, analysis of covariance and effect size for differences in means (partial eta-squared) as a function of the experimental and control groups at pre-test and post-test.

Measures	PRE-TEST					POST-TEST				
	EXPERIMENTAL	CONTROL				EXPERIMENTAL	CONTROL			
	*Mean (SD)*	*Mean (SD)*	*F*	*p*	*µ* ^2^	*Mean (SD)*	*Mean (SD)*	*F*	*p*	*µ* ^2^
**KIDSCREEN**										
HRQL	35.18 (5.84)	35.09 (6.01)	1.414	0.697	0.001	36.94 (6.14)	35.04 (5.99)	1.975	0.018	0.072
**PANASN**										
PA	21.43 (3.90)	20.02 (4.21)	3.293	0.07	0.008	21.95 (2.78)	20.33 (2.88)	5.438	0.017	0.144
NA	11.23 (3.54)	11.58 (3.58)	0.251	0.62	0.004	9.96 (2.95)	11.62 (3.86)	7.044	0.010	0.123
**TEIQUE-ASF**										
TEI	4.82 (0.60)	4.58 (0.64)	4.368	0.04	0.009	5.02 (0.63)	4.52 (0.53)	16.394	0.000	0.241
**SAS**										
SAS-T	2.61 (0.67)	2.64 (0.68)	0.040	0.84	−0.001	2.38 (0.63)	2.58 (0.62)	3.419	0.062	0.014

Note: HRQL = health-related quality of life; PA = positive affect; NA = negative affect; TEI = trait emotional intelligence; SAS-T = Total Social Anxiety Scale; SD = standard deviation.

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
