# Peer review of "Improving Adolescents’ Subjective Well-Being, Trait Emotional Intelligence and Social Anxiety through a Programme Based on the Sport Education Model"

_ijerph, 2019, doi:10.3390/ijerph16101821_

Reviewer 1 Report

The main objective of this study was to assess the impact of a sport education   intervention on subjective well-being, trait emotional intelligence and social anxiety. Although this field of research is very interesting, the paper lacks central methodological and theoretical information and is likely to raise more questions than it answers.

Strengths:
(1) Interesting topic.
(2) Applicability and clinical relevance of results.

Major concerns/limitations study:
(1) Low sample size for this kind of study. In fact, with that sample size It would be necessary to conduct a randomized control trial. In fact, the majority of the psychological assessments were self-reports. Hence, it would be necessary to control that this study was not blinded.
(2) Absence of biological correlates of depression (HPA markers, cortisol immunitarian parameters, etc).
(3) Absence of an appropriate assessment of depression (alternative and validates questionnaires; self-reported symptoms, objective measures of health, etc) in order to control potencial confounding variables.
(4) Absence of an appropriate neuropsychological battery assessment, which would explain the association between those variables.
(5) Explain appropriately clinical applications of the study. 

Author Response

REVIEWER 1

Low sample size for this kind of study. In fact, with that sample size It would be necessary to conduct a randomized control trial. In fact, the majority of the psychological ssessments were self-reports. Hence, it would be necessary to      control that this study was not blinded.

Indeed, we agree that the ideal in research is to use randomised control trials. However, in the educational context it is very difficult to develop this type of design because of the complexity of the organization in the school. In our case, following the researches of other important authors in this context (e.g. Bailey et al., 2009; Siedentop et al., 2019) a random assignment per cluster has been made (that is, some classes groups were randomly assigned to the experimental group and others to the control group). However, sample size has been introduced as a limitation of the study. In addition, the low number of experimental papers with the sport education model (SEM) gives relevance to our pilot study.

We sincerely appreciate the reviewer's contributions. Therefore, in order to respond to the sample suggestions and the use of self-reports as assessment instruments, we have included in the limitations of the study.

Absence of biological correlates of depression (HPA markers, cortisol immunitarian parameters, etc).

We agree with the reviewer that the study of the impact on these biological variables would be very interesting. In this sense, it has been included in future lines of research. However, our study is developed in an educational and non-clinical context.

Absence of an appropriate assessment of depression (alternative and validates questionnaires; self-reported symptoms, objective measures of health, etc) in order to control potential confounding variables.

We appreciate the reviewer's recommendation for depression assessment. In future research, the evaluation of the sports education model (SEM) could focus on this objective. Therefore, this recommendation has been included in future lines of research.

Explain appropriately clinical applications of the study.

Our study was developed in the educational context. In addition, the applications of our work focused on the repercussions on psychosocial adjustment and social and school adaptation.

Reviewer 2 Report

The article analyzes the impact of a physical-sport education pilot programme on adolescents’ subjective well-being, trait emotional intelligence and social anxiety. The results obtained in this study revealed significant improvement in subjective well-being and in trait emotional intelligence in the experimental group after the pilot programme.

Below are general and specific comments to improve the content and readability of the manuscript.

Broad comments:

Although the topic under investigation is of great interest, the sample used is too small for the results to be conclusive.

Specific comments:

7/268: It is necessary to check if the difference between the pretest and posttest of experimental group is statistically significant and also indicate the size of the effect. In the table included this difference is not clear. It is also suggested to carry out a MANOVA of repeated measures taking into account time and group factors.

9/383: It would be necessary to have more updated references, since only 17.5 % of them have been published in the last 5 years.

Author Response

REVIEWER 2

1. Although the topic under investigation is of great interest, the sample used is too small for the results to be conclusive.

We agree with the reviewer that the sample was small and thank him sincerely for his suggestion. Therefore, it has been included in the limitations of the study. However, due to the limited number of experimental investigations with the pedagogical model of sports education, we consider that our results (although preliminary) could generate relevance in educational research.

2. It is necessary to check if the difference between the pretest and posttest of experimental group is statistically significant and also indicate the size of the effect. In the table included this difference is not clear. It is also suggested to carry out a MANOVA of repeated measures taking into account time and group factors.

We appreciate the interesting contribution and suggestion on the inclusion of a MANOVA of repeated measures. Therefore, MANOVA is included between lines 270-272 (p. 7); MANCOVA is included between lines 278-280 (p. 7).

3. It would be necessary to have more updated references, since only 17.5 % of them have been published in the last 5 years.

A few old references have been removed and some more current references have been included.

Reviewer 3 Report

General comments

This article aimed to evaluate the impact of a physical-sport education pilot program on adolescents’ subjective well-being. The main results report a small impact of the program on the dependent variables. In general, the study presents a satisfactory experimental approach to the problem, and by this can be published. However, there are serious flaws regarding a) the logic of the introduction (which is too long and too general), and the interpretation of the results (what impacts on the discussion writing). For this reason, I recommend major revisions.

Specific comments

Introduction

The most important topic of current research is the Sports Education Model. However, from the first to the fourth paragraphs the authors have only described the dependent variables of the study, with no mention to the SEM. I strongly recommend the authors to include information about the pedagogical model since the beginning of the article. The absence of this information, considering the title and the abstract, should confuse the readers.

The introduction is also too long. There is no need to make it so long since the subjects of the article (Sports Education Model and well-being variables) are clearly stated in the literature. In my opinion, seems important to completely reorganize the introduction considering the need to deeply present the research problem instead of widely presenting terms and concepts.

Line 84: “The connection between health and physical activity is widely accepted”. If this connection is already established, what is the novelty of the study? What is the gap addressed by the authors? It is mandatory to deeply present this rationale.

A sports model teaching program is mainly focused on different domains (Metzler, 2011): affective, cognitive and motor. The variables investigated by the authors are more in line with the affective domain, which is, according to the literature, the main purpose of the SEM. So, there is a clear justification for the current study. However, this rationale is not adequately developed through the introduction, since few experimental studies were cited and discussed. The inclusion of more experimental studies regarding the effect of the SEM on participants’ affective domain would increase the deepness of the introduction.

Methods

Why experimental and control groups have a big difference regarding the number of participants? This could bias the results and must be justified by the authors.

Considering that the didactic unity is based on a non-regular sport (Ringo), I recommend the authors to briefly introduce the sport rules and internal logic (is it an invasion or a non-invasion game? What are the most important rules? What are the most important techniques and tactics?). I would also consider to include a representative figure of a Ringo “match”.

What were the criteria for defining the number of sessions within the didactic unity?

How do you assure that the didactic unity really characterized a Sports Education unity? In my opinion, it is mandatory to include a session analysis procedure in order to evaluate if the main principles of the model were followed by the teachers. As an example, see what was conducted in a recent study (Práxedes, Del Villar Álvarez, Moreno, Gil-Arias, & Davids, 2019).

Results

Authors must clearly point out if they are going to consider the differences significant or not. In many cases, there are a) non-relevant effect sizes; and b) non-significant results after Bonferroni’s correction that are assumed as relevant differences later in the discussion topic. In my opinion, in both abovementioned situations, authors must consider that there are no significant differences between and within.

I recommend the authors to include the percentage of change from the first to the second measure, which could indicate the real impact of the instructional program.

Discussion

“The preliminary results obtained in this study revealed significant improvement in subjective well-being in the experimental group after the pilot program”. This result is contrary to what you presented in the results section. The same must be applied to all other variables presented in this paragraph.

Seems like the authors overestimated the impact of the results. In this sense, the discussion must be completely rewritten. Considering what is presented at results section, my general impression is that the SEM has no (or at most limited) impact of the dependent variables, what is contrary to the rationale presented in the discussion.

The same is valid for the conclusion.

References

In my opinion, there is an excessive number of citations that are not relevant to the article. I recommend the authors to include only the most relevant citations. Besides, seems important to include more experimental articles regarding the SEM.

Recommend reading:

Metzler, M. W. (2011). Instructional Models for Physical Education (3rd ed.). Scottsdale, USA: Holcomb Hathaway.

Práxedes, A., Del Villar Álvarez, F., Moreno, A., Gil-Arias, A., & Davids, K. (2019). Effects of a nonlinear pedagogy intervention programme on the emergent tactical behaviours of youth footballers. Physical Education and Sport Pedagogy, 1–12. https://doi.org/10.1080/17408989.2019.1580689

Author Response

REVIEWER 3

The most important topic of current research is the Sports Education Model. However, from the first to the fourth paragraphs the authors have only described the dependent variables of the study, with no mention to the SEM. I strongly recommend the authors to include information about the pedagogical model since the beginning of the article. The absence of this information, considering the title and the abstract, should confuse the readers.

We appreciate the interesting contribution and suggestion cited. Therefore, the requested change has been applied.

The introduction is also too long. There is no need to make it so long since the subjects of the article (Sports Education Model and well-being variables) are clearly stated in the literature. In my opinion, seems important to completely reorganize the introduction considering the need to deeply present the research problem instead of widely presenting terms and concepts.

We appreciate the interesting contribution and suggestion cited. Therefore, the requested change has been applied.

“The connection between health and physical activity is widely accepted”. If this connection is already established, what is the novelty of the study? What is the gap addressed by the authors? It is mandatory to deeply present this rationale.

A more extensive justification has been applied in lines 43-46 (p.2).

A sports model teaching program is mainly focused on different domains (Metzler, 2011): affective, cognitive and motor. The variables investigated by the authors are more in line with the affective domain, which is, according to the literature, the main purpose of the SEM. So, there is a clear justification for the current study. However, this rationale is not adequately developed through the introduction, since few experimental studies were cited and discussed. The inclusion of more experimental studies regarding the effect of the SEM on participants’ affective domain would increase the deepness of the introduction.

We appreciate the reviewer's suggestion. Thus, suggested reference has been included in the reference number [16].

Why experimental and control groups have a big difference regarding the number of participants? This could bias the results and must be justified by the authors.

When developing cluster random trial, it was impossible for half of one class group to participate in the intervention and the other not. The educational centre had to have teachers who attended half of the group-class that did not participate and that was impossible.

Considering that the didactic unity is based on a non-regular sport (Ringo), I recommend the authors to briefly introduce the sport rules and internal logic (is it an invasion or a non-invasion game? What are the most important rules? What are the most important techniques and tactics?). I would also consider to include a representative figure of a Ringo “match”.

A short ringo explanation has been introduced in lines 201-203 (p.5). In addition, an explanatory figure (Figure 1) is added (p.6/lines 218-220).

What were the criteria for defining the number of sessions within the didactic unity?

The criteria for defining the number of sessions were: (1) Long-term didactic units favour team affiliation or belonging (Siedentop et al. 2019); (2) the recommendations of the Department of Didactics of Physical Education in the educational centre.

How do you assure that the didactic unity really characterized a Sports Education unity? In my opinion, it is mandatory to include a session analysis procedure in order to evaluate if the main principles of the model were followed by the teachers. As an example, see what was conducted in a recent study (Práxedes, Del Villar Álvarez, Moreno, Gil-Arias, & Davids, 2019).

We appreciate and share your contribution. In this sense, we have adopted the recommendations of the sport education model (SEM) established by Hastie and Casey’s guidelines were followed for the design and validation of the programme [19] (p. 423): a) thoroughly detailed curricular elements; b) precise certification of the applied model; and c) an in-depth explanation of the context of the programme. The intervention programme was implemented in the experimental group following sequencing of content and activities in three stages (initial, intermediate and final) over 16 sessions (Table 2).

We consider that we have been very exhaustive (designers, applicators, evaluators and researchers) in the design and development of the sessions according to the SEM. However, to respond to this recommendation, the reference provided by the reviewer is included (p.9/line 350-351) (Reference No. [75]: Práxedes, A.; Del Villar Álvarez, F.; Moreno, A.; Gil-Arias, A.; Davids, K. Effects of a nonlinear pedagogy intervention programme on the emergent tactical behaviours of youth footballers. Phys. Educ. Sport Peda. 2019, 1-12, DOI: 10.1080/17408989.2019.1580689).

Authors must clearly point out if they are going to consider the differences significant or not. In many cases, there are a) non-relevant effect sizes; and b) non-significant results after Bonferroni’s correction that are assumed as relevant differences later in the discussion topic. In my opinion, in both abovementioned situations, authors must consider that there are no significant differences between and within.

The requested change has been applied (Results).

Seems like the authors overestimated the impact of the results. In this sense, the discussion must be completely rewritten. Considering what is presented at results section, my general impression is that the SEM has no (or at most limited) impact of the dependent variables, what is contrary to the rationale presented in the discussion.

The requested change has been applied.

The same is valid for the conclusion.

The requested change has been applied.

In my opinion, there is an excessive number of citations that are not relevant to the article. I recommend the authors to include only the most relevant citations. Besides, seems important to include more experimental articles regarding the SEM.

The requested change has been applied.

Recommend reading:
A few references have been updated and removed. In addition, the requested references have been included:

Metzler, M. W. (2011). Instructional Models for Physical Education (3rd ed.). Scottsdale, USA: Holcomb Hathaway.

Práxedes, A., Del Villar Álvarez, F., Moreno, A., Gil-Arias, A., & Davids, K. (2019). Effects of a nonlinear pedagogy intervention programme on the emergent tactical behaviours of youth footballers. Physical Education and Sport Pedagogy, 1–12. https://doi.org/10.1080/17408989.2019.1580689.

Round  2

Reviewer 1 Report

Despite my initial objections to accept this manuscript, it seems that the editor is very interested in it. In addition, the authors have worked hard to improve the article. Therefore, the only recommendation that suggest to the authors is that they should add the cronbach alpha in the questionnaires that they employed for this study.

Author Response

Therefore, the only recommendation that suggest to the authors is that they should add the cronbach alpha in the questionnaires that they employed for this study.

The requested revision is included in Table 3. In the lines 266-267

Reviewer 2 Report

The suggestions made have been taken into account by the authors. We agree with the changes made.

Author Response

Many thanks to the reviewers for their work.

Best regards.

This manuscript is a resubmission of an earlier submission. The following is a list of the peer review reports and author responses from that submission.

Round  1

Reviewer 1 Report

This manuscript aimed to study whether  physical-sport programme might interfere with self-reported well-being, trait emotional intelligence and social anxiety. Although interesting, there are several problems with this manuscript, which I discuss below and it is essential to solve: 

1) It would be necessary to conduct a randomized control trial. In fact, the majority of the psychological assessments were self-reports. Hence, it would be necessary to control that this study was not blinded.

2) Is the data normally distributed?

3) I think that you really need to apply Bonferroni corrections for multiple comparisons.

4) It should be necessary to moderate your conclusions based on your design study.

Reviewer 2 Report

This is a very promising area of research.  The article could be revised for submission but I’m not sure that it will meet IJERPH standards.  It may be better for the authors to aim for a lower level journal and perhaps present this work as a pilot investigation.  This is a study in a single school and it is important to be conservative about the conclusions from the research.

The introduction is too general.  Health is defined in the opening paragraph and the next paragraph is about societal goals for wellbeing.  The introduction didn’t seem to pin down the specifics such as previous studies of sports education as an intervention and interventions to improve well-being, emotional intelligence and social anxiety.  These points are definitely mentioned, but there isn’t enough detail to see how the present study relates to previous research or whether there is a clear rationale for the present study. 

For the sample, I didn’t understand what was meant by ‘natural groups’.  I didn’t see any attempt to check if the sample size was adequate for the outcome measures. Because this is a non-clinical sample, demonstrating changes in e.g. the SAS-T are difficult and there is a risk of both type 1 and type 2 errors if the sample size is too small.  Was there a reason for having more participants in the experimental group than in the control group? If so, it should be explained.  I wondered about contamination from experimental to control given that the children were from the same school – was there any mechanism to prevent this from happening? 

There wasn’t enough detail about the intervention.  For example, it wasn’t clear who was involved in instruction – it would be helpful to know how many staff were involved and their qualification level.  Sessions 1-3 seemed to be about team identity rather than PE.  Sessions 4-7 – I wasn’t sure how much was practical and how much was theoretical.  In general, it was difficult to understand the intervention. 

Some of the measures used have been validated in Spanish, but not all measures.  What procedures where used to ensure the other measures were suitable for use with Spanish children?

ANOVA was used, but there isn’t enough details in the reporting.  Was this a repeated measures ANOVA?  Conventions for reporting ANOVA should be followed and for ANCOVA the covariates should be clear.  Eta squared is usually reported for effect size with repeated measures ANOVA and is included in the SPSS output (I note the authors used SPSS).  I wondered why the authors had selected Cohen’s d instead – noting that I have no objection to use of Cohen’s d, but this probably needs explanation.

I checked the scoring of the SAS-T as used in the source referred to (reference 69) and they added the scores for each of the items rather than taking the mean score for all of the items.  I’m not sure that this would make a difference in terms of results, but it is important to ensure scoring is done in a way that is consistent with previous studies.

Last, I also wondered how the intervention differed from the general experiences of the participants.  The intervention seemed to involve the types of activities that any child involved in sport would have had exposure to.  Interventions typically work because something new is being introduced, but perhaps the new element was related to identity building or another attentional factor rather than sports related factors.

The manuscript is difficult to read because of excessive use of acronyms.  It’s important to remember that unfamiliar acronyms reduce the speed of reading due to the extra cognitive effort of retrieving the information.  Acronyms often look good to the writer but are rarely welcomed by the reader. 

Avoid causal language.  Even with a well-powered RCT, researchers are generally use caution regarding causation.  In this article, it is stated in the abstract that the “programme caused significant improvements” and there are similar statements/suggestions elsewhere.

As mentioned at earlier, there is a lot of promise in this research and it would be great to see a larger study in the future.